# Shadowing and shielding: Effective heuristics for continuous influence maximisation in the voting dynamics

**Guillermo Romero Moreno** *, Sukankana Chakraborty, Markus Brede

School of Electronics and Computer Science, University of Southampton, Southampton, United Kingdom

* E-mail: Guillermo.RomeroMoreno@soton.ac.uk

## Abstract

Influence maximisation, or how to affect the intrinsic opinion dynamics of a social group, is relevant for many applications, such as information campaigns, political competition, or marketing. Previous literature on influence maximisation has mostly explored discrete allocations of influence, i.e. optimally choosing a finite fixed number of nodes to target. Here, we study the generalised problem of continuous influence maximisation where nodes can be targeted with flexible intensity. We focus on optimal influence allocations against a passive opponent and compare the structure of the solutions in the continuous and discrete regimes. We find that, whereas hub allocations play a central role in explaining optimal allocations in the discrete regime, their explanatory power is strongly reduced in the continuous regime. Instead, we find that optimal continuous strategies are very well described by two other patterns: (i) targeting the same nodes as the opponent (shadowing) and (ii) targeting direct neighbours of the opponent (shielding). Finally, we investigate the game-theoretic scenario of two active opponents and show that the unique pure Nash equilibrium is to target all nodes equally. These results expose fundamental differences in the solutions to discrete and continuous regimes and provide novel effective heuristics for continuous influence maximisation.

## Introduction

Due to the unprecedented interconnectedness achieved in current societies [1, 2], the study of opinion dynamics has been a topic that has found much attention in recent years. The abundance and immediacy of virtual communication have allowed an almost instantaneous spread of information, leading to new phenomena of much interest for both researchers and the broader public, such as echo chambers [3], viral posts [4, 5], and opinion polarisation [6]. One important problem in this field is influence maximisation, which is relevant because of its links to social policies [7, 8], politics [9], or advertising [10]. Influence maximisation (IM) focuses on how external forces can exploit the opinion dynamics of a social group to maximise the spread of the desired opinion [10, 11]. Researchers in the field chiefly address the question of which individuals should be influenced—under a budget constraint—to achieve maximum

from the website of the University Rovira i Virgili
(https://deim.urv.cat/~alexandre.arenas/data/
xarxes/email.zip).

**Funding:** MB acknowledges support from the Alan
Turing Institute (EPSRC grant EP/N510129/1,
https://www.turing.ac.uk/) and the Royal Society
(grant IES\R2\192206, https://royalsociety.org/).
SC is sponsored by the U.S. Army Research
Laboratory and the U.K. Ministry of Defence under
Agreement Number W911NF-16-3-0001. The
views and conclusions contained in this document
are those of the authors and should not be
interpreted as representing the official policies,
either expressed or implied, of the U.S. Army
Research Laboratory, the U.S. Government, the U.
K. Ministry of Defence or the U.K. Government. The
U.S. and U.K. Governments are authorized to
reproduce and distribute reprints for Government
purposes notwithstanding any copyright notation
hereon. The funders had no role in study design,
data collection and analysis, decision to publish, or
preparation of the manuscript.

**Competing interests:** The authors have declared
that no competing interests exist.

spread. Studies on IM allow us to better understand different aspects of influencing attempts, such as under which conditions one can effectively introduce the desired opinion in a population [12, 13], how influencing strategies interact when competition for influence takes place [14, 15], or what the effect of external influence is on reaching consensus in a population [16].

Modelling the spread of opinions has found much attention in the past and many different models have been proposed to capture various aspects of opinion dynamics [17, 18]. *Static* models (also known as *progressive* models) have an established tradition in the IM literature [11, 19], particularly featuring the independent cascade [20] or threshold [21] models. Models in this class only allow opinion changes in one direction and typically model the spread of a novel opinion brought into a social group by a small group of individuals. However, this modelling assumption only applies to short timescales and is not suitable for changing environments [22], so a significant amount of IM literature has instead employed *dynamic* models [23–26] which attempt to describe the stochastic fluctuations of opinions over time.

Among dynamic models, the voter model [27, 28] stands out for its long tradition in the field and its mathematical tractability. In the voter model, opinion dynamics are driven by individuals copying the opinions of peers they interact with. Despite the simplicity of this update rule, the model has been found to match very well with real data at the aggregate level [29, 30]. Although in the voter model opinions flicker back and forth, homogeneous populations eventually often arrive at a consensus in timescales that depend on the network topology [31]. However, this situation changes in the presence of agents with fixed opinions, commonly known as *zealots* [22, 32, 33], *stubborn agents* [23, 24, 34], or *frozen nodes* [35, 36], which drive the population to a dynamic equilibrium with both opinions present [32]. Due to the important role that zealots play in the composition of opinions in equilibrium, some works conceive IM in the voter model as the optimal placement of zealots in the network [23, 24]. This approach assumes that nodes can be readily transformed into zealots of the desired opinions without considering the costs and effort of conversion, which is too strong an assumption. In contrast, other approaches more related to network control, model IM via external zealots that choose how to exert influence on a limited set of nodes in the social network [26, 33].

Previous research on IM —both on static and dynamic models— has mainly employed a discrete approach, i.e. identifying a set of *K* individuals whose control maximises the share of the desired opinion [7, 8, 10, 11, 22, 23, 26, 33, 37–41]. However, the discretisation of influence targeting is artificially constraining if campaigners wish to allocate different amounts of resources to different groups. The latter strategy can be better captured by a continuous approach where the external controller can tailor the intensity of influence to individual target nodes [42–44].

Importantly, there are significant differences in optimal strategies in the discrete and continuous allocation regimes regarding the structure of optimal solutions. For instance, targeting hub nodes or nodes with high centrality are common heuristics that have been shown to perform well in the discrete allocation regime [22, 33, 37–40, 45, 46]. While other studies have demonstrated that taking into account the full topology performs better than such heuristics, they mainly provide algorithmic approximations to optimal solutions without examining details of the topology of resultant optimal influence allocations [8, 11, 41]. In contrast, the focus of the present paper is on providing analytical solutions and explaining optimal allocations. In the continuous allocation regime, previous work of us for the voter model points to an absence of preference towards hub nodes, while highlighting instead the dependence on the strategy of an opponent for choosing optimal allocations [43, 44, 47]. We presented two simple heuristics that are effective for IM: a direct response to the nodes targeted by the opponent (termed *shadowing*) and an indirect response that focuses on the neighbours of the nodes targeted by the opponent (termed *shielding*). We introduced them through numerical examples

and provided a deeper analysis on star and K-regular graphs, but analytical results and analysis on general complex network topologies is still outstanding and will be presented in this paper.

Here, we combine and extend our previous work to achieve a complete characterisation of the structure of optimal allocation strategies. We achieve this by providing four main contributions. First, we develop a more general framework for understanding continuous IM for the voter model on complex networks. By combining analytical results and numerical techniques, we analyse the structure of optimal solutions and establish that they tend to be spread across large parts of the network. Second, by comparing the structure of optimal solutions in the discrete and continuous allocation regimes, we provide a deeper understanding of continuous optimal allocation and explain it through various heuristics. More specifically, we explore to what extent optimal solutions in the continuous and discrete regimes can be explained by hub targeting, shadowing, or shielding. Our results demonstrate that whereas hub preferences are very important in the discrete setting, shadowing and shielding heuristics are the predominant factors explaining maximal influence in the continuous case. Third, we quantify the degree to which an opponent strategy can be exploited by an active controller. Our result shows that strategies are the more exploitable, the more their allocations to low–degree nodes deviate from optimality. Fourth, we analyse the game–theoretical scenario where both external controllers actively adapt their strategy and provide empirical understanding of the Nash equilibrium.

## Model of opinion dynamics and framework for IM

In the following, we consider a group of $N$ agents represented by the nodes of a social network. Social connections are given by positively weighted links of strength $w_{ij}$. Agents are assumed to hold one of two possible binary opinions ($A$ or $B$) and opinion diffusion follows the classical voter model [27, 28], with nodes copying the opinions of neighbours with a probability proportional to the weight of the link connecting them. Further to the standard model, external influence is introduced via zealots as external controllers [26, 33, 48]. We assume there are two external controllers respectively holding either of the two opinions. These controllers exert influence on the network via unidirectional links with weights $w_{ai}$ ($w_{bi}$) that are also taken into account in the updating dynamics of opinions. In the continuous regime, controllers can freely decide which nodes they target, with positive continuous strengths $w_{ai}, w_{bi} \in \mathbb{R}^+$ and subject to a budget constraint, $\mathcal{B} \geq \sum_i w_{ai} (w_{bi})$. The discrete regime is a sub-case of the continuous regime which only allows two possible weights, $w_{ai}, w_{bi} \in \{0, g\}$, where $g$ is a fixed *gain*. The number $K$ of nodes thus targeted is related to the budget via $K = \lfloor \mathcal{B}/g \rfloor$.

The behaviour of the opinion dynamics can be studied by considering the evolution of probabilities $x_i(t) \in [0, 1]$ that a node holds opinion $A$ [32, 33, 40]. From this perspective, the dynamics of a node can be described by the rate equation

$$\frac{dx_i}{dt} = (1 - x_i)\frac{\sum_j w_{ij}x_j + w_{ai}}{d_i + w_{ai} + w_{bi}} - x_i\frac{\sum_j w_{ij}(1 - x_j) + w_{bi}}{d_i + w_{ai} + w_{bi}}, \qquad (1)$$

where $d_i$ is the weighted degree of a node, $d_i = \sum_j w_{ij}$.

We focus on IM in the steady-state, i.e. when the system has arrived at its unique attractor $\boldsymbol{x}^*$, given by $(L + W_a + W_b)\boldsymbol{x}^* = \boldsymbol{w}_a$[33], where $L$ is the weighted Laplacian of the network, $W_a$ ($W_b$) is a diagonal matrix whose diagonal entries correspond to $w_{ai}$ ($w_{bi}$), and bold symbols are column vectors. The total vote share of nodes holding opinion $A$ at the equilibrium is obtained

from

$$X = \frac{1}{N}\sum_i x_i = \frac{1}{N}\mathbf{1}^T \mathbf{x}^* = \frac{1}{N}\mathbf{1}^T (L + W_a + W_b)^{-1} W_a \mathbf{1} \ . \tag{2}$$

For the IM problem, without loss of generality, we assume that the controller favouring $A$ is active, i.e. she seeks to find the best way to distribute her link weights, $\mathbf{w}_a$, with an aim to maximise her vote share, $X$. We generally assume that the opposing controller (favouring $B$) is passive, with fixed links $\mathbf{w}_b$ known by the active controller, and that controllers have full information about the network structure. Formally,

$$\max_{W_a} X, \ \ s.t. \ \ \ X = \frac{1}{N}\mathbf{1}^T (L + W_a + W_b)^{-1} W_a \mathbf{1}, \ \ \ \sum_i w_{ai} \le \mathcal{B}_a, \ \ w_{ai} \ge 0 \ . \tag{3}$$

The continuous modelling of IM brings a methodological benefit: the assumption of continuous strengths allows for better mathematical treatment of the problem since local search techniques can be applied for finding an optimum, as opposed to the combinatorial optimisation that is required in the discrete IM problem. Moreover, as the IM problem defined in our framework is concave (for the proof, see Section 12 in S1 Appendix), convex optimisation techniques such as gradient ascent are guaranteed to obtain arbitrarily accurate solutions.

## Results

Our results are structured as follows. We first perform an initial exploration of the structure of optimal allocation strategies in the continuous allocation regime. Second, we investigate in more depth the three heuristics that can be used for better understanding optimal allocations in the continuous regime: shadowing, shielding, and hub preferences. For this purpose, we provide an analytical approximation to optimal allocations that explains shadowing for arbitrary network topologies and further quantify the extent to which shadowing and shielding can account for optimal influence configurations. Third, we perform a similar analysis on optimal solutions in the discrete regime and directly compare the contribution of shadowing, shielding, and hub targeting heuristics in either regime. Last, we look at the game-theoretical scenario and characterise the Nash equilibrium of two active opponents.

For illustration, the experiments shown in the manuscript are performed on the real–world email interaction network from [49] of size $N = 1133$, as an example of a typical topology found in social networks. Details about the network can be found in Section 1 in S1 Appendix and the network data can be found in S1 Dataset. However, the results shown here also apply to more general classes of complex networks, as it is demonstrated in Section 11 in S1 Appendix. Optimal continuous and discrete allocations are numerically determined via gradient ascent and stochastic hill climbing for the continuous and discrete regimes, respectively. See the Methods section for more details.

### Structure of optimal allocations in the continuous regime

We first qualitatively explore the shape of optimal allocations in the continuous regime. The continuous regime allows for richer allocation decisions as controllers can reach all nodes in the network and modulate the strength of the allocation given to each node. If linked to practical examples, this would correspond to campaign managers using diverse modes of campaigning, potentially mixing a soft campaign directed to all individuals in a population with stronger campaigning directed at specific groups. Here, we address the question, if campaign managers would benefit from using such diverse modes of campaigning. For an initial

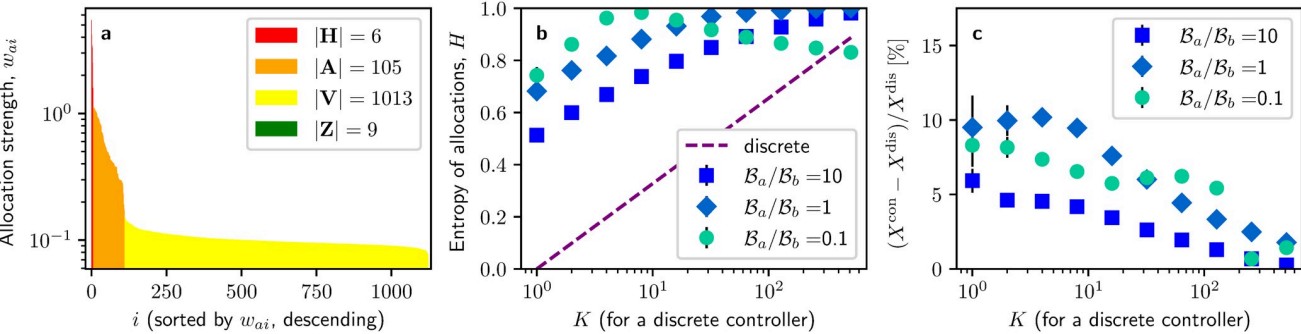

**Fig 1. Preliminary exploration of the optimal influence profile in the continuous regime. a** Distribution of optimal influence allocations sorted in descending order when the passive controller discretely targets $K = 16$ randomly chosen nodes and both controllers hold the same budget, $\mathcal{B} = N\langle d\rangle/60$. Allocations are coloured by the allocation group they belong to. **b** Entropy of optimal allocation distributions in response to passive controllers with discrete allocations that are more or less widely spread (as measured by the number of nodes $K$ that they target) and different budget ratios. The entropy of discrete allocations of the passive controller is also depicted (*dashed line*). **c** Percentage enhancement of control by optimal continuous allocations over optimal discrete allocations for varying $K$ and budget ratios. Error bars represent standard errors over 15 experiment samples and the total budget of both controllers sums up to $\mathcal{B}_a + \mathcal{B}_b = N\langle d\rangle/30$.

exploration, we set the passive controller to target $K$ random nodes in the network with equal strength. We then numerically compute optimal allocations for the active controller via gradient ascent for continuous allocations and hill–climbing for discrete allocations (see the Methods section for more details).

In Fig 1a, we present optimal control allocations sorted in descending order and obtained via gradient ascent. Results shown in the figure have been obtained from a numerical experiment where the passive controller targets $K = 16$ nodes and both controllers have equal budgets. To facilitate the analysis and description of the optimal allocation profile, we have divided allocations into disjoint groups by targeting strength $w_{ai}$ with respect to the mean allocation $\langle \boldsymbol{w}_a\rangle$. We have thus distinguished four disjoint groups $\mathbf{G} = \{\mathbf{H}, \mathbf{A}, \mathbf{V}, \mathbf{Z}\}$ of nodes with high allocations $\mathbf{H} = \{i|w_{ai} \geq 8\langle \boldsymbol{w}_a\rangle\}$, above-average allocations $\mathbf{A} = \{i|8\langle \boldsymbol{w}_a\rangle > w_{ai} \geq \langle \boldsymbol{w}_a\rangle\}$, below-average allocations $\mathbf{V} = \{i|\langle \boldsymbol{w}_a\rangle > w_{ai} > 0\}$, and zero allocations $\mathbf{Z} = \{i|w_{ai} = 0\}$. Note that we have chosen this particular partitioning to characterise the four distinct groups that can be observed in the optimal allocation profile from Fig 1a and that this grouping is mainly employed for illustrative purposes.

If a controller targeted $K$ nodes in a discrete fashion (and $K < N/8$), allocations to the $K$ targeted nodes would belong to the high-allocation group $\mathbf{H}$ while all remaining allocations would belong to the zero-allocation group $\mathbf{Z}$. In contrast, we consider that a controller makes use of the continuous flexibility if allocations are spread across the network (with few nodes belonging to $\mathbf{Z}$) and cover a wide range of strengths. Fig 1a illustrates that even though there is a focus of heavy targeting on some selected nodes ($|\mathbf{H}| + |\mathbf{A}| = 111$) and some nodes remain untargeted ($|\mathbf{Z}| = 9$), the allocation profile of the remaining nodes ($|\mathbf{V}| = 1013$) is mostly flat, with little variation of allocations around their mean, $std(\boldsymbol{w}_a^V)/\langle \boldsymbol{w}_a^V\rangle = 0.11$. From these results, we see that the active controller is making use of the continuous flexibility, as allocations are widely spread across the network (only $|\mathbf{Z}| = 7$ nodes did not receive allocation at all) and across strengths (ranging from $w_{ai} = 10^{-1}$ to $w_{ai} = 5$). Optimal allocations for other values of $K$ can be seen in Fig 2 in S1 Appendix.

To extend the analysis of the allocation profile to other scenarios, we use the entropy of the allocation distribution as a characterisation of the spread of optimal allocations, $H(\boldsymbol{w}_a) = -\log(N)\sum_i w_{ai}/\mathcal{B}_a \log(w_{ai}/\mathcal{B}_a)$. Note that this entropy is normalised to the interval $[0, 1]$, which is achieved by the pre-factor $log(N)$. Controllers that make use of the targeting

flexibility of the continuous regime will tend to spread their allocations across the network, resulting in higher entropies. Fig 1**b** shows that, almost independent of $K$ and budget ratios, distributions of continuous allocations have considerably larger entropy than those of their discrete counterparts. We further note that the degree of spread is affected by the sparseness of the passive controller: the larger $K$ and thus the more widely spread the strategy of the passive controller is, the more wide-spread optimal allocations are as well. In contrast, relative budgets ratios do not have a major effect on the entropy of the allocation distribution.

We next evaluate possible improvements obtainable by campaigning with continuous influence allocations, as measured by the relative improvement in vote shares gained by optimal continuous allocations over optimal discrete allocations, $(X^{cont} - X^{disc})/X^{disc}$. In Fig 1**c**, we see that relative improvements are generally in the range of 5–10% against passive controllers with $K < 100$, while relative improvements drop below 5% against wide-spread passive controllers with $K > 100$. Note that, although the improvements can be small, they may be of significant importance in winner–takes–all scenarios, such as political elections or referendums. Extensions to other network topologies of experiments from Fig 1**a** and 1**c** can be seen in Fig 10 in S1 Appendix, showing similar results.

To summarise, we have seen that spreading allocations over the whole network brings a bigger benefit to an optimal controller than when concentrating them on a few nodes, even if these are hubs. We also see that spreading allocations widely is optimal regardless of how many nodes are targeted by the opponent or the budget ratio between controllers. However, we also note that in optimal configurations some nodes are targeted much more strongly than others, with a particular focus on fairly few selected nodes $|\mathbf{H}| + |\mathbf{A}| = 111$, which comprise around 10% of the network. We next explore how these nodes are chosen, i.e. if they can be related to any of the shadowing, shielding, or hub preference heuristics.

## Shadowing, shielding, and hub preferences in the continuous regime

To proceed, we investigate the explanatory power of shadowing, shielding, and node degree heuristics for determining optimal above-average allocations in the continuous regime. While node degree is related to the hub-targeting heuristics commonly found in the discrete regime, here we extend earlier work of us [43, 44] and propose shadowing and shielding as effective heuristics for influence maximisation. We first note that the shadowing and shielding heuristics can be classified by an order, given by the distance of node targeting from nodes targeted by the opponent. In this sense, shadowing can be considered a first-order strategy, as it directly affects nodes targeted by the opponent. In contrast, shielding constitutes a second-order heuristic, as it concerns nodes in the direct environment of nodes targeted by the opponent. Further-order shielding is also possible, although not explored in depth here due to its higher complexity and lower effect.

**First-order optimal responses: Shadowing or avoidance.** In this subsection, we aim to develop analytical intuition behind the shadowing behaviour for general network topologies. Due to the analytical intractability of the IM problem on complex networks, we use a heterogeneous mean-field (HMF) approximation to the opinion dynamics. In this approximation, we assume that, independent of the details of individual neighbourhoods, every node is coupled to a mean-field. The expression for the vote share obtained in this approximation is given by

$$X^{\text{HMF}} = \frac{1}{N}\left(\sum_i \frac{d_i}{d_i + w_{ai} + w_{bi}}\right)\left(\sum_i \frac{d_i w_{ai}}{d_i + w_{ai} + w_{bi}}\right)\left(\sum_i \frac{d_i(w_{ai} + w_{bi})}{d_i + w_{ai} + w_{bi}}\right)^{-1} +$$
$$+ \frac{1}{N}\sum_i \frac{w_{ai}}{d_i + w_{ai} + w_{bi}} \ . \tag{4}$$

(Refer to the Methods section for more details about the approximation and the derivation of Eq (4).) Eq (4) is non-linear in $w_{ai}$, so finding analytical expressions for optima is extremely challenging. Therefore, we focus on the limiting cases of small and large external allocations with respect to the connectivity of the network to get analytical solutions. We thus perform series expansions in the limits of $(w_{ai} + w_{bi}) \ll d_i$ and $(w_{ai} + w_{bi}) \gg d_i$, which allow obtaining closed-form solutions for optimal strategies. (Refer to Sections 2 and 3 in S1 Appendix for details about their derivations).

We start with the low allocations limit $(w_{ai} + w_{bi}) \ll d_i$, for which we obtain optimal allocations $w_{ai}^L$ as

$$w_{ai}^L = \frac{\mathcal{B}_a + \mathcal{B}_b}{2N} + \frac{1}{2}\left(\frac{\mathcal{B}_a}{\mathcal{B}_b} - 1\right)\left[w_{bi} + \frac{d_i}{\sum_{j:w_{aj}^L>0} d_j}\left(\mathcal{B}_b - \sum_{j:w_{aj}^L>0} w_{bj}\right)\right], \tag{5}$$

where the sums over $\sum_{j:w_{aj}^L>0}$ are an effect of the positivity constraint $w_{ai} \geq 0$. From Eq (5), we can observe that optimal allocations $w_{ai}^L$ depend linearly on allocations of the opponent $w_{bi}$ and node degree $d_i$, both terms multiplied by a common coefficient whose sign depends on the ratio of budgets: $c = (\mathcal{B}_a/\mathcal{B}_b - 1)$. If $\mathcal{B}_a > \mathcal{B}_b$, i.e. if the active controller is in advantage of resources, the coefficient $c$ is positive and the active controller allocates more influence to the nodes that are strongly targeted by the opponent, which corresponds to the shadowing strategy as discussed above. On the contrary, if $\mathcal{B}_a < \mathcal{B}_b$, the coefficient $c$ is negative and the active controller allocates more influence to the nodes that receive the least allocation by the opponent, corresponding to avoidance behaviour. Note that, when the positivity constraints are not active ($w_{ai}^L > 0$, $\forall i$), there is no dependence of optimal allocations $w_{ai}^L$ on node degree $d_i$, as $(\mathcal{B}_b - \sum_{j:w_{aj}>0} w_{bj}) = 0$. In such cases, optimal allocations only depend on the allocations of the opponent $w_{bi}$ and the budgets $\mathcal{B}_a$ and $\mathcal{B}_b$. Since the allocations of the opponent cannot be negative either, the positivity constraints of the active controller can only be activated when $\mathcal{B}_a < \mathcal{B}_b$. If positivity constraints are activated, optimal allocations also depend linearly on node degree $d_i$ with a negative coefficient $c$, giving more allocations to nodes with low degree. Hence, there is no scenario in which this analytical solution favours high-degree nodes.

The expression of the total vote share $X^L$ for this limiting case given a passive controller with allocations $\boldsymbol{w}_b$ and an optimal active controller is

$$X^L = \frac{\mathcal{B}_a}{\mathcal{B}_a + \mathcal{B}_b} + \frac{\mathcal{B}_b}{4N^2}\sum_i \frac{(1 - w_{bi}/\langle \boldsymbol{w}_b \rangle)^2}{d_i}, \tag{6}$$

where $\langle \boldsymbol{w}_b \rangle = \mathcal{B}_b/N$. We note that the vote share $X^L$ obtainable for the active controller is the higher the more the allocations of the passive controller $w_{bi}$ deviate from the mean $\langle \boldsymbol{w}_b \rangle$. In other words, the passive controller is the more exploitable by an active controller the more her passive strategy deviates from uniform targeting. Interestingly, the impact of this deviation is the stronger the lower the degree $d_i$ of the node mis-targeted; thus controllers are the more exploitable the less carefully they allocate resources to low-degree nodes. We confirm this result via numerical experiments (as can be seen in Fig 3 in S1 Appendix).

In the opposite limit of large allocations with respect to node degree, $(w_{ai} + w_{bi}) \gg d_i$, we have optimal allocations $w_{ai}^H$ as

$$w_{ai}^H = \left(\mathcal{B}_a + \sum_{j:w_{aj}^H>0} w_{bj}\right) \frac{\sqrt{w_{bi}}}{\sum_{j:w_{aj}^H>0}\sqrt{w_{bj}}} - w_{bi}. \tag{7}$$

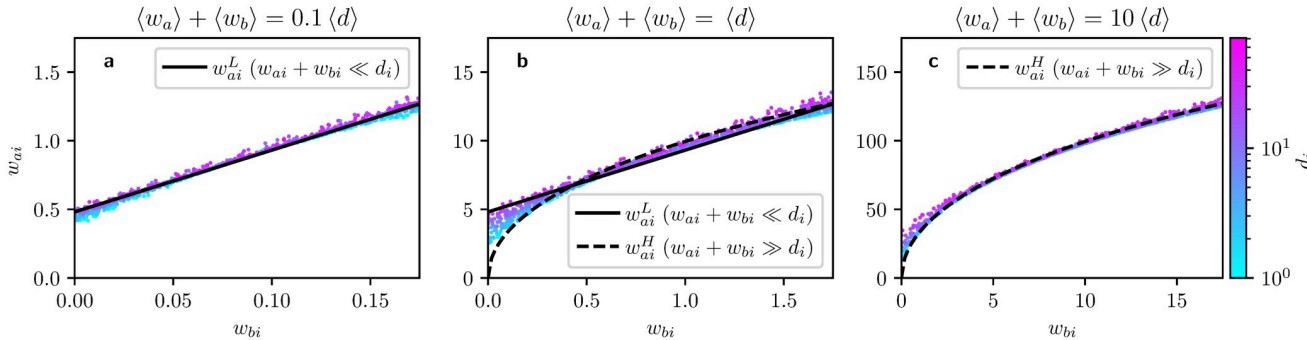

**Fig 2. Comparison of analytical and numerical results for optimal influence allocations on the continuous regime.** Each panel show the dependence of optimal allocations $w_{ai}$ on the allocations of a passive controller $w_{bi}$ when the value of external allocations are generally **a** lower, **b** equal, or **c** higher than node degrees, as shown on the top of each panel. The active controller is in budget superiority ($\mathcal{B}_a/\mathcal{B}_b = 10$) and the passive controller targets continuously with weights randomly drawn from a uniform distribution. Numerical results are given by a cloud of points resolved by classes of nodes of different degrees, indicated by their colour, and analytical results are given by *black curves* that correspond to Eq (5) (*solid*) or Eq (7) (*dashed*).

In this limit, optimal allocations are only dependent on the allocations of the opponent $\boldsymbol{w}_b$ in a non-linear fashion and the available budget $\mathcal{B}_a$. This solution does not show any dependence on node degree $d_i$. The expression of the vote share $X^H$ for this limiting case with an optimal active controller against a passive controller is

$$X^H = 1 - \frac{\mathcal{B}_b}{\mathcal{B}_a + \mathcal{B}_b} \left( \frac{1}{N} \sum_i \sqrt{w_{bi}/\langle \boldsymbol{w}_b \rangle} \right)^2 . \tag{8}$$

Again, we find that the vote share $X^H$ is the higher the more individual node allocations of the passive controller $w_{bi}$ deviate from the mean $\langle \boldsymbol{w}_b \rangle$. In this limit we find no effect of node degree $d_i$, i.e. controllers are equally exploitable independent of on which nodes they misallocate their resources.

To verify the validity of the assumptions involved in the above approximation, we compare the analytical optimal allocations to optimal allocations obtained via numerical experiments in different scenarios. To cover a wide range of $w_{bi}$, we set the passive controller to target nodes with strengths $w_{bi}$ randomly drawn from a uniform distribution. In Fig 2, we examine the dependence of optimal responses $w_{ai}$ on allocations of the passive controller $w_{bi}$ for scenarios in which the active controller is in budget superiority and with allocations generally being **a** smaller, **b** similar, or **c** larger than node degree. In the figure, results corresponding to the HMF-approximation are given as black curves, while numerical results are given as clouds of points.

We observe that the agreement between numerical and analytical results is very good in the limiting cases, cf. Fig 2**a** and 2**c**. When allocations are similar in magnitude to node degree (Fig 2**b**), numerical results are bounded by the analytical results for both limiting cases. Another set of experiments where the active controller is in budget inferiority can be found in Section 6 in S1 Appendix. Those experiments show cases where the positivity constraints are active, so for the low-allocation limit the term that depends on $d_i$ activates. Analytical and numerical results also match reasonably well in the budget inferiority scenario.

**Second-order optimal responses: Shielding.** As we have seen above, first-order responses can explain optimal allocations against opponents who spread their influence widely across the network. However, as the HMF does not distinguish node classes depending on the distance to

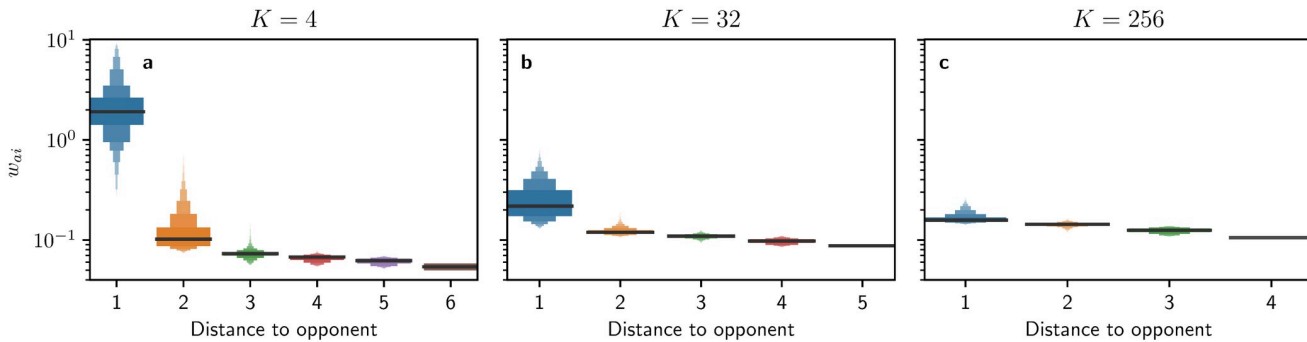

**Fig 3. Depiction of the shielding heuristic.** On the y axis, numerical optimal control allocations of the active controller in the form of a letter-value plot [50]. On the x-axis, shortest distance to a node targeted by the passive controller who targets **a** $K = 4$, **b** $K = 32$, or **c** $K = 256$ random nodes in the network in a discrete fashion. Different levels of a boxplot represent the median, quartiles, octiles, and so on, while containing points from 15 repeats of the experiment. The total budget of both controllers sums up to $\mathcal{B}_a + \mathcal{B}_b = N\langle d\rangle/30$.

the targets of the opponent, a first-order heuristic is likely to be insufficient when opponents only target relatively few nodes in the network. In such cases, we need to resort to second-order responses (i.e. shielding) for explaining optimal allocations. The idea behind shielding is that nodes that are direct neighbours of the nodes targeted by the opponent should receive higher allocations than other nodes in the network. The presence of shielding is then indicated by a strong dependence of allocations to nodes on the distance to nodes targeted by the passive controller, which becomes evident from the numerical experiments shown in Fig 3. We also observe that shielding decreases in importance the more nodes $K$ the passive controller targets, as seen in Fig 3c with $K = 256$, where differences in allocations barely vary with the distance.

Once we have assessed that shielding is present in optimal allocations, we now turn back to the allocation groups in Fig 1 and ask the question, to which extent shadowing and shielding can explain the particular partitioning **G**. For this purpose, we define a new set of disjoint groups related to shielding $\mathbf{S} = \{\mathbf{T}_b, \mathbf{N}_b, \mathbf{R}\}$ with nodes that are targeted by the passive controller ($\mathbf{T}_b = \{i|w_{bi} = g\}$), direct neighbours of nodes targeted by the passive controller ($\mathbf{N}_b = \{i \notin \mathbf{T}_b| (\exists j \in \mathbf{T}_b|w_{ij} > 0)\}$) and the remaining nodes ($\mathbf{R} = \{i \notin (\mathbf{T}_b \cup \mathbf{N}_b)\}$). Fig 4a shows the overlap between the shielding groups in **S** and the allocation groups in **G** for the case where the passive

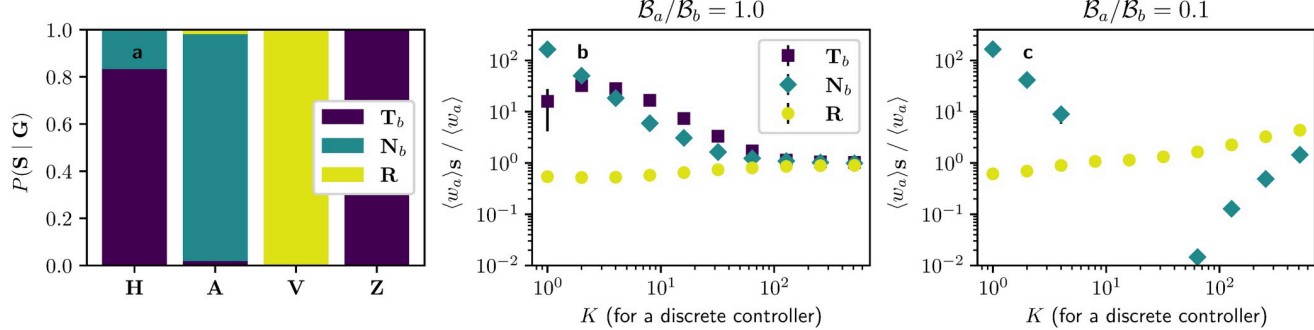

**Fig 4. Analysis of the shielding heuristic on optimal influence allocations in the continuous regime. a** Overlap between the allocation groups in **G** and shielding groups in **S** for a passive controller targeting $K = 16$ nodes in a discrete fashion. **b, c** Average allocations given to each group in **S** with respect to the total average allocation $\langle w_{ai}\rangle$ for different number of nodes $K$ targeted by the passive controller and for **b** equal budgets or **c** budget disadvantage. Error bars in **b, c** represent standard errors calculated from 15 repeats of the experiment. Points for $\mathbf{N}_b$ in the interval $K \in [5, 35]$ do not appear in **c** since their value is $\langle w_a\rangle_{N_b} = 0$ and the scale is logarithmic. The total budget of both controllers sums up to $\mathcal{B}_a + \mathcal{B}_b = N\langle d\rangle/30$.

controller targets $K = 16$ random nodes in the network. We observe that nodes that receive very high (**H**) or zero (**Z**) allocation are mainly nodes targeted by the passive controller ($\mathbf{T}_b$), the nodes that receive above-average allocations (**A**) are mainly the neighbours of the nodes targeted by the passive controller ($\mathbf{N}_b$), and the nodes that receive below-average allocations (**V**) are the remaining ones (**R**). This result demonstrates that the different allocations given to nodes in the network are largely explained by their distance to the node targeted by the opponent.

We extend the analysis beyond allocation groups in **G** and to other values of $K$ by exploring average allocations $\langle w_a \rangle$ given to the shielding groups in **S** (c.f. Fig 4b and 4c). When in budget equality (Fig 4b), we observe that average allocations $\langle w_a \rangle$ given to nodes in $\mathbf{T}_b$ and $\mathbf{N}_b$ are much larger than those given to nodes in **R** for $K < 50$, although this difference diminishes for higher $K$. These results show that both shielding and shadowing have a significant presence for low $K$. However, it is worth recalling from Fig 4a that allocations given to nodes in $\mathbf{T}_b$ receive either high or zero allocations, so its distribution is bimodal and not well captured by the mean value. Extensions to other network topologies of experiments from Fig 4b can be seen in Fig 10 in S1 Appendix, pointing to analogous results. Fig 4c shows the same scenarios as in Fig 4b but with the active controller in budget disadvantage. Here, nodes in $\mathbf{T}_b$ receive zero allocation, corresponding to the avoidance behaviour discussed above. Interestingly, in this scenario nodes in $\mathbf{N}_b$ also receive less allocation $\langle w_a \rangle$ than those in **R** for $K > 5$, so avoidance also appears as a second-order strategy when in budget disadvantage. So direct (first-order) avoidance of the nodes targeted by the opponent is complemented by a second-order avoid-ance (*anti-shielding*) provided the budget disadvantage is large enough. The case of budget advantage is not shown here, as it is very similar to that of budget equality, only with larger allocations given to nodes in $\mathbf{T}_b$.

**Hub preferences and dependence on node degree.** The third heuristic that we investigate is hub-targeting or, more generally, a degree–preference heuristic. We have seen that shadow-ing and shielding can explain much of optimal influence strategies in the continuous regime. However, we note that the effects of shielding and node degree are related, as nodes with high degree are more likely to be neighbours of nodes targeted by the opponent and hence should also receive higher allocations due to shielding. Can a degree–preference heuristic explain optimal allocations more accurately than shielding or is the degree–preference just a heuristic that approximates the effect of shielding?

To test whether the shielding and degree–preference strategies are related, we compare the conditional probabilities of a node belonging to the above-average optimal allocation group **A** given node degree $P(i \in \mathbf{A}|d_i)$, with the conditional probabilities of a node being neighbour of a node targeted by the opponent given node degree $P(i \in \mathbf{N}_b|d_i)$, as shown in Fig 5a. We find that the probability profiles are very similar between the two groups, so it is very likely that one of these two heuristics is an artefact of the other one and thus can be superseded.

In the following, by measuring Kendall rank correlation coefficients $\tau$ [51] between target-ing strengths $w_{ai}$ and node degree $d_i$, we extend the above analysis to scenarios with other $K$. A preference towards hubs would be indicated by correlations close to $\tau = 1$. Fig 5b and 5c com-pare correlations $\tau$ between the optimal strategy (*squares*) and a strategy purely based on shielding (*circles*), i.e. equally dividing all the resources among nodes in $\mathbf{N}_b$. In Fig 5b, we observe that both the optimal and shielding strategies show considerable correlations with node degree for $K > 30$, where correlations are generally above $\tau > 0.5$. The lack of correla-tions with node degree for low $K$ and the similarities in correlations of the optimal and shield-ing strategies indicates that a degree–preference heuristic is ineffective unless shielding is correlated to node degree. To further illustrate this point, we perform another set of experi-ments in which we ensure that shielding and node degree are not positively correlated. We

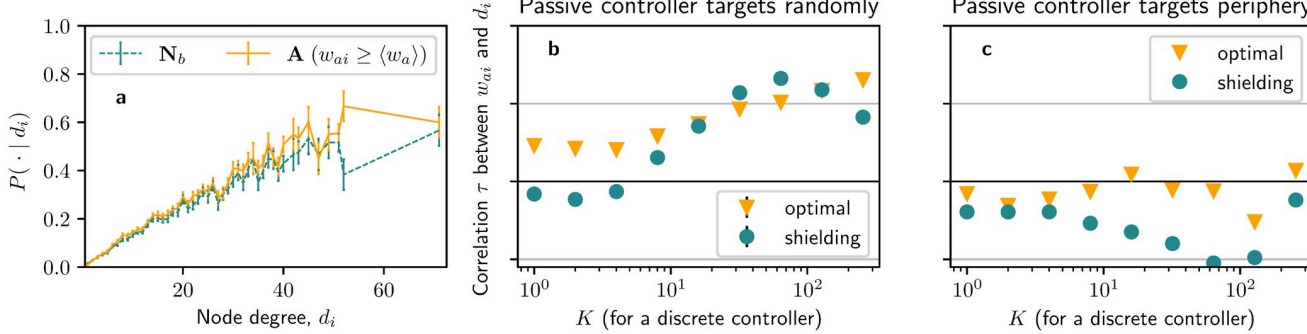

**Fig 5. Correlations between optimal influence allocations and node degree. a** Probability of optimal allocations belonging to **A** given node degree $d_i$ (*solid*) and probability of nodes belonging to $\mathbf{N}_b$ given node degree $d_i$ (*dashed*), with a passive controller targeting $K = 16$ random nodes in a discrete fashion. **b, c** Kendall rank correlation coefficients $\tau$ between allocation strength $w_{ai}$ and node degree $d_i$ for different $K$ when the active controller targets nodes optimally (*squares*) or following a shielding strategy (*circles*). In **a** and **b** the passive controller targets random nodes in the network. In **c**, the passive controller targets the nodes whose neighbours have the lowest possible degree. Error bars represent standard errors over **a** 60 or **b** 15 instances of the experiments. Correlations in **c** correspond to a single instance of the experiment, as the strategy of the passive controller is deterministic. The total budget of both controllers sums up to $\mathcal{B}_a + \mathcal{B}_b = N\langle d \rangle / 30$.

achieve this by having the passive controller only targeting nodes whose neighbours have the smallest possible degree in the network, thus effectively displacing the targets of shielding to nodes with low degree. Indeed, as shown in Fig 5c, the correlations of optimal allocations with node degree vanish in these scenarios, with values below $|\tau| < 0.25$, while correlations of the shielding strategy remain negative with values down to $\tau \approx -0.5$. Extensions to other network topologies of experiments from Fig 5b can be seen in Fig 10 in S1 Appendix, showing similar results. Also, a plot directly comparing node degrees $d_i$ and optimal allocations $w_{ai}$ can be found in Fig 5 in S1 Appendix.

In conclusion, we note a mild preference towards hubs in the continuous regime, as high-degree nodes have higher probabilities of receiving higher allocations than other nodes. However, we also argued that these correlations with node degree are related to the shielding heuristics since nodes with a high degree have higher chances of being neighbours of nodes targeted by the opponent. When we investigate scenarios with low $K$ or with the passive controller targeting peripheral nodes whose neighbours have a low degree, we observe that the correlation with node degree vanishes, while being similar to that obtained by a shielding strategy. Therefore, we conclude that the correlations with node degree seen here are an artefact of the shielding strategy. Optimal allocations can be explained almost perfectly by the shielding heuristics without the need for additional assumptions about degree preference.

## Comparison of heuristics for optimal configurations in the continuous and discrete regimes

In the previous sections, we have obtained optimal allocation strategies via numerical methods and have analysed the structure of these solutions while comparing them with some reference strategies. From these analyses, we have evaluated the importance of the three proposed explanatory factors (degree preference, shadowing, and shielding) in the continuous IM by studying their presence in optimal allocations. In this subsection, we complement the above results with an inverse approach: we build a set of heuristics based on the three factors and measure the gap in vote share $\Delta X$ relative to optimal results. The smaller the gap $\Delta X$ of a heuristic to the numerically determined optimum, the stronger its contribution to optimal

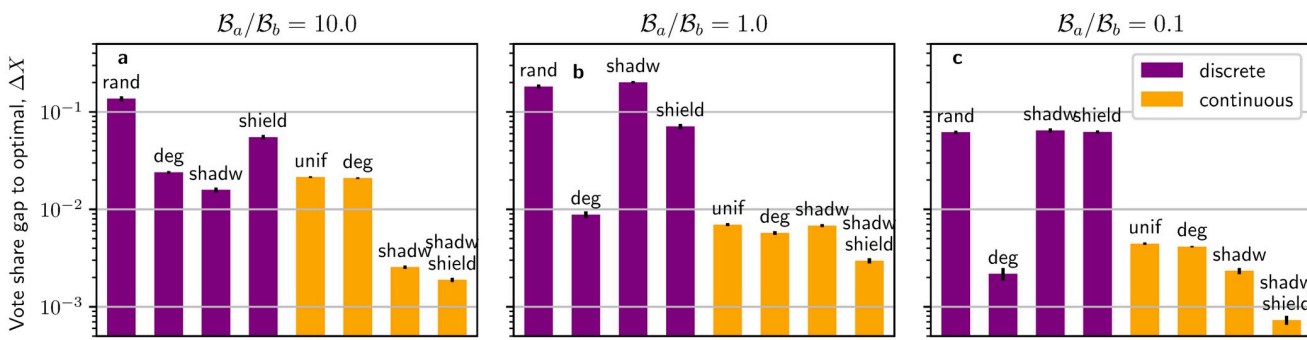

**Fig 6. Comparison of various heuristics to optimal allocations.** Bars represent the gap in vote share $\Delta X$ of the heuristics with respect to optimal numerical allocations for three different budget scenarios (as indicated on the top of the panels). Each bar represents one of the following heuristics: random (*rand*), degree-based (*deg*), shadowing-based (*shadw*), shielding-based (*shield*), uniform targeting (*unif*), combination of shadowing and shielding (*shadw shield*). The passive controller targets $K = 16$ nodes in the network in a discrete fashion. Error bars represent standard errors for 15 instances of the experiments and the total budget of both controllers sums up to $\mathcal{B}_a + \mathcal{B}_b = N\langle d \rangle/30$.

allocations. Additionally, we also extend the current study to the discrete regime and compare how the three proposed heuristics perform when the active controller is constrained to discrete allocations. A more detailed investigation on the role of the three heuristics in optimal allocations for the discrete regime can be found in Section 8 in S1 Appendix.

Fig 6 illustrates results of the vote share gap for experiments run for three budget scenarios (advantage, equality, and disadvantage) and against a discrete passive controller who targets $K = 16$ random nodes in the network. Note that we make a clear separation between the discrete (*purple*) and continuous (*orange*) regimes and that their vote shares are compared to the discrete or continuous optimal strategies, respectively. The details of the implementation of each heuristic can be found in the Methods section below.

From Fig 6, we make the following observations about the three budget scenarios. First, when in budget advantage (Fig 6a), we see that shadowing is the best-performing discrete heuristics with a gap of $\Delta X = 1.58 \cdot 10^{-2}$, closely followed by the degree-based heuristic with $\Delta X = 2.41 \cdot 10^{-2}$. Among the continuous heuristics, the uniform ($\Delta X = 2.14 \cdot 10^{-2}$) and degree-based ($\Delta X = 2.09 \cdot 10^{-2}$) heuristics perform clearly worse than shadowing ($\Delta X = 2.55 \cdot 10^{-3}$) and shadowing-plus-shielding ($\Delta X = 1.89 \cdot 10^{-3}$). We can conclude that when in budget advantage, shadowing is an important driver in both the discrete and continuous regimes. Second, for budget equality (Fig 6b), the degree-based heuristic performs the closest to the optimal discrete strategy ($\Delta X = 8.99 \cdot 10^{-3}$). Among the continuous heuristics, shadowing-plus-shielding is again most effective ($\Delta X = 2.97 \cdot 10^{-3}$), with the other three heuristics within the range $\Delta X \in [5.5 \cdot 10^{-3}, 7.0 \cdot 10^{-3}]$. We see that when in budget equality, node degree is a very important factor in the discrete regime, while shadowing-plus-shielding is most effective in the continuous regime. Last, when in budget disadvantage (Fig 6c), the discrete degree-based heuristic ($\Delta X = 2.12 \cdot 10^{-3}$) again proves superior to the other heuristics, which perform similar to random allocations. Among the continuous heuristics, shadowing-plus-shielding is again performing best ($\Delta X = 7.28 \cdot 10^{-4}$), with the other three performing in the range $\Delta X \in [2.33 \cdot 10^{-3}, 4.44 \cdot 10^{-3}]$. In this scenario, the degree-based heuristic is again a very important factor in the discrete regime, while shadowing-plus-shielding is predominant in the continuous regime. An alternative assessment of the heuristics based on computational efforts is shown in Fig 14 in S1 Appendix.

In summary, in the discrete regime node degree is consistently found to be the main driver for optimal allocations, which is only slightly surpassed by shadowing when in budget

advantage. In contrast, in the continuous regime shadowing and shielding have a considerably stronger effect than node degree across all scenarios. This effect is particularly pronounced when the active controller is in budget advantage (where shadowing is very effective) or in budget disadvantage (where shielding is dominant). Extensions to other network topologies of experiments from Fig 6 can be found in Figs 12 and 13 in S1 Appendix, showing similar results.

### Nash equilibrium on a game with two active controllers

Up to this point, we have regarded the opponent as passive, i.e. with a fixed strategy that is known by the active controller. In this subsection, we investigate the case where both controllers actively optimise their strategy. We assume that they simultaneously choose their strategy without having any information about the opponent. Since optimal strategies are subject to the opponent's decisions, we are interested here in finding the pair of strategies that are simultaneously optimal for both controllers —the pure-strategy *Nash equilibrium* [52]—, so neither of them is willing to deviate from it.

Due to the concavity of our problem with respect to either controller's strategy, it can be proven that there is a unique pure-strategy Nash equilibrium (for the proof, see Section 12 in S1 Appendix). We can numerically arrive at the Nash equilibrium by employing gradient-ascent to iteratively eliminate dominated strategies until convergence to the pure-strategy Nash Equilibrium. We find that in the Nash Equilibrium, independent of the network type and budgets, both controllers uniformly target all nodes equally. In the numerical results, this is seen by the convergence of the normalised standard deviation of control allocations, which (up to numerical precision) converges to zero (Fig 7 in S1 Appendix illustrates such behaviour for two different budget ratios).

Uniform targeting at the Nash equilibrium can also be interpreted from the perspective of the shadowing and shielding heuristics. On the one hand, we can also find the Nash equilibrium analytically from the first-order approximations derived above. Based on Eqs (5) and (7) and exploiting the symmetry in exchanging the A and B controllers, we can find mutually optimal responses as $w_{ai} = \mathcal{B}_a/N$, $w_{bi} = \mathcal{B}_b/N$, i.e. uniform targeting. On the other hand, the shielding heuristics gains more advantage when opponents concentrate their influence on a few nodes, as shown in Fig 4, so spreading allocations across the network avoids being exploited by shielding.

## Discussion

In this paper, we have investigated how continuous and discrete allocation regimes affect the structure of optimal allocations in influence maximisation for the voter dynamics. Our focus is on scenarios with two opposing controllers, one passive and one actively optimising, for which we have defined the continuous influence maximisation problem. As an initial result, we have seen that the continuous regime adds some improvement in the objective function, which can be linked to the benefits of spreading control allocations across the whole network. Regarding the structure of optimal allocations, we have pointed to three important heuristics that can be used for its characterisation: shadowing, shielding, and degree dependence. Despite the three heuristics playing a role in explaining optimal allocations in both the continuous and discrete regimes, their relative contributions vary. While optimal allocations in the discrete regime are mainly driven by node degree, the shielding and shadowing heuristics are predominant explanatory factors in the continuous regime. In the game-theoretical scenario where both controllers simultaneously optimise their influence allocations, the pure-strategy Nash equilibrium in the continuous regime is assumed when both controllers uniformly target all nodes in

the network —regardless of details of the network topology. Our results mainly apply to the scenario in which a controller reacts against a passive opponent. However, note that this framework can be mapped to the scenario in which a single controller attempts to influence a population that manifests some resistance to the external influence, as shown in [26]. In particular, such resistance to external control is equivalent to a passive opponent who targets nodes with a strength proportional to the degree of the node, so a degree–dependent strategy becomes much more predominant when only one external controller is featured.

This paper contributes to the general understanding of controlling dynamical processes on networks [53] in three ways. First, it provides three explanatory factors as tools that can be employed to characterise any solution to a control problem: degree dependence, shadowing, and shielding. Second, it shows that, in the particular case of the voter model with external controllers, the degree-dependence factor is the main driver of controlling solutions in the discrete regime, while the shadowing-plus-shielding combination is the main explanatory factor behind solutions in the continuous regime. These results can be used for guidelines as to what to expect in other scenarios. Lastly, the proposed factors can be employed as heuristics in cases where optimal solutions cannot easily be obtained numerically, or alternatively as benchmarks for testing approximately optimal solutions.

Some of our results relate to other findings from previous literature. Most works refer to a preference towards hubs when linking optimal solutions to a degree dependence. However, we have shown that correlations with node degree can also be negative (i.e. targeting peripheral nodes), particularly when the controller is in budget disadvantage). Brede, Restocci and Stein [26, 48, 54] also show that the preference towards hub nodes shifts to a preference towards periphery nodes in the discrete regime, although under other conditions; namely when optimising for short time horizons [54] or in the presence of noise in the opinion dynamics [26, 48]. They also suggest that for short time horizons the optimal strategy is also opponent-dependent, shadowing nodes with initially opposing opinions if they are few and avoiding them when they are many [54]. Our results have pointed to a lack of degree dependence in the continuous case. This independence is not fully reflected in the studies on the continuous IM on the Ising model with noise by Lynn and Lee [25, 41]. Unlike in our results, they find that optimal targets are also related to node degree in the continuous regime, as periphery nodes should be targeted when temperatures are high and hub nodes for intermediate temperatures. However, when the temperature of the system is very low (i.e. nodes are very susceptible to the external influence), a uniform allocation of influence is the best strategy against an also uniform opponent, and hence there is no correlation with node degree in the absence of noise [25]. In light of this result, studying the inclusion of noise would be a natural extension to this work, as it could also generate a degree dependence in the continuous IM for the voter dynamics.

There are many other ways in which this work can be extended. As mentioned above, we have mainly focused on the differences in optimal strategies between the continuous and discrete regimes in the voter model. The current analysis can also be extended to other models of opinion dynamics that allow for continuous allocations, such as the Ising model [55], or the epidemic-related SIR [56]. Likewise, we use in our experiments a heterogeneous network that is generally well mixed (without strong communities) and has very little degree assortativity (i.e. correlations in node degree across links). It would be of interest to study the impact of shadowing and shielding in these other settings. A further limitation of our work is that we have only considered strategies that do not vary over time. However, campaigners in more realistic settings might vary their allocation at different stages of the dynamics. For an extension to influence maximisation with dynamic allocations, see [54, 57].

In conclusion, we have highlighted the impact that having a continuous or discrete regime may have on both the quality and structure of optimal solutions to network control. Furthermore, we have proposed three factors that characterise solutions to control diffusion problems where opposing external controls influence the network: shadowing, shielding, and degree dependence. The application of this framework to other contexts may facilitate comparisons across different modelling decisions and problems.

## Methods

### Solving the IM problem in the continuous regime

A first step towards solving the continuous IM optimisation problem is finding the critical points where the gradient of vote share $X$ with respect to control allocations $\boldsymbol{w}_a$ is zero,

$$\nabla_{\boldsymbol{w}_a} X = \frac{1}{N} [\mathbf{1}^T \cdot (L + W_a + W_b)^{-1} [I - \mathrm{diag}(x_i^*)]]^T = \mathbf{0} \,, \tag{9}$$

where $\mathrm{diag}(x_i^*)$ is a diagonal matrix whose diagonal elements correspond to the probability vector at the steady state $\boldsymbol{x}^* = (L + W_a + W_b)^{-1} W_a \mathbf{1}$.

Since $\nabla_{\boldsymbol{w}_a} X = \mathbf{0}$ is generally nonlinear in $\boldsymbol{w}_a$—and analytically intractable for the general case—we propose two different approaches to finding a solution: analytically studying a heterogeneous mean-field approximation, which simplifies the calculations under some assumptions, and a numerical solution via gradient ascent for the general case.

### Heterogeneous mean-field approximation

The heterogeneous mean-field (HMF) approximation assumes that, independent of the details of individual neighbourhoods, every node is coupled to a mean-field where the probability of having a node as a neighbour is proportional to their degree [34, 48, 58, 59]. This approximation is not always valid but tends to perform well for not overly sparse networks in the absence of particular degree correlations. The HMF approximation assumes random connections between nodes and ignores higher-order correlations that might be present in real-world networks.

Under this approximation, the expected behaviour of a neighbour $\langle x \rangle$ is presumed to be the same for every node in the network and equal to

$$\langle x \rangle = \frac{1}{N} \sum_i \frac{d_i}{\langle d \rangle} x_i \,, \tag{10}$$

Note that high-degree nodes are more likely to be found as neighbours and hence the product by $d_i/\langle d \rangle$. We thus obtain probabilities of adopting A in the steady state as

$$x_i^* = \frac{d_i \langle x^* \rangle + w_{ai}}{d_i + w_{ai} + w_{bi}} . \tag{11}$$

But Eq (11) can be re-inserted into Eq (10), arriving at a self-consistency equation and leading to the form-closed expression of the vote share as

$$X^{\mathrm{HMF}} = \frac{1}{N} \left( \sum_i \frac{d_i}{d_i + w_{ai} + w_{bi}} \right) \left( \sum_i \frac{d_i w_{ai}}{d_i + w_{ai} + w_{bi}} \right) \left( \sum_i \frac{d_i (w_{ai} + w_{bi})}{d_i + w_{ai} + w_{bi}} \right)^{-1} + $$
$$+ \frac{1}{N} \sum_i \frac{w_{ai}}{d_i + w_{ai} + w_{bi}} \,. \tag{12}$$

## Gradient ascent algorithm

Due to the concavity of our IM problem in the strategy space (for the proof, see Section 12 in S1 Appendix), local-search techniques are guaranteed to arrive at the global optimum [60]. We employ here gradient ascent as a local-search method (Algorithm 1), which has previously been used in IM [44, 61]. This technique is ensured to reach an $\epsilon$-approximation to the exact solution in $O(1/\epsilon)$ iterations of the algorithm [60]. Solutions must meet the budget and positivity constraints, so after every iteration they are projected back to the regular $N$-simplex that fulfils $\sum_i w_{ai} = \mathcal{B}_a$, $w_{ai} \geq 0$.

**Algorithm 1**: $\epsilon$-approximation to optimal allocation of influence via gradient ascent

```
input: ℬₐ, L, Wᵦ, μ, ϵ
output: approximation for wₐ* at the global maximum, X*
1  wₐᵢᵗ⁼⁰ = ℬₐ/N;   Xᵗ⁼⁰ = 0;
2  repeat
3      ∇_wₐ Xᵗ⁻¹ = (9);
4      wₐ' = wₐᵗ⁻¹ + μ∇_wₐ Xᵗ⁻¹;
5      wₐᵗ = Projection of wₐ' onto the N-simplex constraint;
6      Xᵗ = (2);
7  until Xᵗ − Xᵗ⁻¹ < ϵ
```

The parameter $\mu$ (line 4 of Algorithm 1) regulates the step size of the gradient ascent method. Big steps will generally speed up the process but may risk its convergence. To prevent divergence, *backtracking* is employed after each iteration; i.e. the new solution is rejected and the step size halved if the vote share is lower in the new iteration than in the previous one. The projection to the N-simplex is done following the algorithm developed in [62]. Note that every step requires a $N \times N$ matrix inversion, so the time complexity to reach an $\epsilon$-approximate solution scales with $N$ and $\epsilon$ as $O(N^3/\epsilon)$.

For our experiments, we have used $\mu = 50$, $\epsilon = 10^{-10}$. All solutions converged in less than $10^5$ iterations.

## Solving IM in the discrete regime: Hill–climbing algorithm

Solving IM in the discrete regime requires combinatorial optimisation, as the problem is reduced to finding the subset of $K$ nodes that maximises the vote share in the equilibrium [11]. Due to our relatively large problem size ($N = 1133$), searching for the exact maximum would be highly costly even for subset sizes as small as $K < 5$. We hence resort to a local-search technique, the stochastic hill–climbing, which has been previously used for solving discrete IM in the voter model [26, 33, 48]. The hill–climbing algorithm departs from a random subset of $K$ nodes and iteratively swaps one of them with a random node in the network. The modified subset is preserved after each iteration if it performs better than the previous subset; otherwise, the change is reversed. The algorithm typically runs for a given number of iterations or until there has been a specific number of iterations without improvement. This algorithm is only guaranteed to reach the global optimum in convex problems, so in non-convex problems a few runs from different initial positions are desirable. Note that the expected number of iterations needed to find the last optimal node is $K(N − K)$, i.e. $K$ iterations to pick the last non-optimal node from the subset times $(N − K)$ iterations to swap it to its optimal position.

For all experiments in this paper, hill–climbing is performed for $20N$ iterations or until no change is made in the last $10N$ iterations, whichever occurs last.

## Heuristics

Next, we explain in more detail how each of the heuristics from Fig 6 is implemented.

- *rand (discrete), random*: Random targeting of $K$ nodes in the network with equal strength.

- *deg (discrete), degree–based strategy*: This discrete heuristics targets the $K$ nodes that have either the highest or the lowest degree in the network, whichever results in the highest vote share.

- *shadw (discrete), shadowing*: Discrete shadowing targets exactly the same nodes as the passive controller.

- *shield (discrete), shielding*: Discrete shielding targets direct neighbours of the nodes targeted by the passive controller, giving preference to nodes that are neighbours of more than one node. If $K$ is larger than the size of the neighbourhood $|\mathbf{N}_b|$, the remaining targets are directed to random nodes not targeted by the passive controller.

- *unif (continuous), uniform*: Targeting all nodes in the network with equal strength $w_{ai} = \mathcal{B}_a/N$.

- *deg (continuous), degree–based strategy*: Targeting nodes proportional to their degree. The proportionality constant is found numerically via binary search. Note that *unif* is a sub-case of this heuristic in which the proportionality constant is equal to zero, so this heuristic will always perform equal or better than *unif*.

- *shadw (continuous), shadowing*: Continuous shadowing where nodes are targeted with two possible intensities, one for the nodes targeted by the passive controller and another one for the remaining nodes. The values of these intensities are determined numerically via binary search. Note that *unif* is a sub-case of this heuristic in which both intensities are equal, so this heuristic will always perform equal or better than *unif*.

- *shadw shield (continuous), shadowing plus shielding*: Combination of shadowing and shielding where nodes are targeted with three possible intensities depending on whether they are i) targeted by the passive controller ($\mathbf{T}_b$), ii) their neighbours ($\mathbf{N}_b$), or iii) the remaining nodes ($\mathbf{R}$). The values of these intensities are determined numerically via binary search. Note that *shadw (continuous)* is a sub-case of this heuristic in which the allocations to nodes in $\mathbf{N}_b$ and $\mathbf{R}$ are equal, so this heuristic will always perform equal or better than *shadw (continuous)* (and consequently than *unif*).

Note that the proposed heuristics require different amounts of information for their implementation. Some require the details of the network structure and/or knowledge about the opponent's strategy. Others have free parameters and their implementation requires an exploration of the parameter space, with computations of expected vote shares for evaluating each value of the parameters. Table 1 summarises what information is required for each of the heuristics.

**Table 1. Information required by each heuristic.**

|  | Network structure | Opponent's strategy | Parameter Tuning |
|---|:---:|:---:|:---:|
| Discrete heuristics |  |  |  |
| *random* | - | - | - |
| *degree–based* | X | - | X |
| *shadowing* | - | X | - |
| *shielding* | X | X | - |
| Continuous heuristics |  |  |  |
| *uniform* | - | - | - |
| *degree–based* | X | - | X |
| *shadowing* | - | X | X |
| *shadowing plus shielding* | X | X | X |

## Supporting information

**S1 Appendix. Appendix with extra derivations and experiments as supporting information.**
(PDF)

**S1 Dataset. List of edges of the network of e-mail interchanges between members of the University Rovira i Virgili (Tarragona) [49].** This dataset has been retrieved from https://deim.urv.cat/~alexandre.arenas/data/xarxes/email.zip.
(TXT)

## Acknowledgments

The authors acknowledge the use of the IRIDIS High Performance Computing Facility, and associated support services at the University of Southampton, in the completion of this work.

## Author Contributions

**Conceptualization:** Guillermo Romero Moreno, Sukankana Chakraborty, Markus Brede.

**Formal analysis:** Guillermo Romero Moreno, Sukankana Chakraborty.

**Investigation:** Guillermo Romero Moreno, Markus Brede.

**Methodology:** Guillermo Romero Moreno, Sukankana Chakraborty, Markus Brede.

**Software:** Guillermo Romero Moreno, Sukankana Chakraborty.

**Supervision:** Markus Brede.

**Validation:** Markus Brede.

**Visualization:** Guillermo Romero Moreno.

**Writing – original draft:** Guillermo Romero Moreno, Sukankana Chakraborty.

**Writing – review & editing:** Guillermo Romero Moreno, Markus Brede.

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
