## [Decision Letter · Decision Letter 0]

3 Mar 2021

PONE-D-21-02851

Shadowing and shielding: Effective heuristics for continuous influence maximisation in the voting dynamics

PLOS ONE

Dear Dr. Romero Moreno,

Thank you for submitting your manuscript to PLOS ONE. After careful consideration, we feel that it has merit but does not fully meet PLOS ONE’s publication criteria as it currently stands. Therefore, we invite you to submit a revised version of the manuscript that addresses the points raised during the review process.

Following the referees advices the paper should be ready for publication in next round.

We look forward to receiving your revised manuscript.

Kind regards,

Hocine Cherifi

Academic Editor

PLOS ONE

Journal Requirements:

Reviewers' comments:

Reviewer's Responses to Questions

**Comments to the Author**

1. Is the manuscript technically sound, and do the data support the conclusions?

Reviewer #1: Yes

Reviewer #2: Yes

2. Has the statistical analysis been performed appropriately and rigorously? 

Reviewer #1: Yes

Reviewer #2: Yes

3. Have the authors made all data underlying the findings in their manuscript fully available?

Reviewer #1: Yes

Reviewer #2: Yes

4. Is the manuscript presented in an intelligible fashion and written in standard English?

Reviewer #1: Yes

Reviewer #2: Yes

5. Review Comments to the Author

Reviewer #1: In this paper, the authors address the question of how to maximize influence in the context of the voter model with opinion leaders. I very much enjoyed reading the paper. It is clearly written, and the idea of integrating control theory with models of opinion dynamics is interesting and important. In particular, the demonstration that the ‘shadowing’ and ‘shielding’ strategies are two ends of a spectrum, which depends on the relative strength between the external influence weights and the node degree, is a nice result. My intuition, however, tells me that the optimal allocation for each node should closely correspond to the weighted degree of the node. The authors need to conduct further experiments to convince me otherwise. I have included below several comments that need to be addressed by the authors:

* Comment 1. The authors need to provide additional details regarding the experiments in Figure 1. What is the size of the network? How are the weights (i.e., wij) determined in this experiment?

* Comment 2. Regarding the optimal control allocation (not ‘shadowing’ and ‘shielding’), the authors need to include a plot relating the weighted degrees of nodes (i.e., di) to their corresponding optimal allocations (i.e., w_ai). What do you get?

* Comment 3. What is the distribution of the optimal allocation values (Figure 1)? When changing the model’s parameters (K, B, wij, network structure), do you see any interesting scaling regimes related to the optimal allocation distribution? How does the allocation distribution compare with the weighted degree distribution?

* Comment 4. In your model you assume that the control vector is independent of time. In real life scenarios (e.g. in the context of political campaigns) it is likely that campaigners change their allocation based on real-time information on the current vote share of nodes in the system (e.g. as estimated by polls). It would be nice to comment on this possible extension of your model.

* Comment 5. Perhaps I am missing something, but don’t you have an unneeded “N” in your equation of the entropy (on Page 5)?

* Comment 6. The following paper, I believe, is one of the first papers that introduced the concept of “zealots” in the context of opinion dynamics (along with Mobilia’s paper):

Chinellato, D. D., de Aguiar, M. A., Epstein, I. R., Braha, D., & Bar-Yam, Y. (2007). Dynamical response of networks under external perturbations: exact results. arXiv preprint arXiv:0705.4607.

An extension of this paper also appeared in:

Chinellato, D. D., Epstein, I. R., Braha, D., Bar-Yam, Y., & de Aguiar, M. A. (2015). Dynamical response of networks under external perturbations: exact results. Journal of Statistical Physics, 159(2), 221-230.

It would be appropriate to include them in your references.

Reviewer #2: he authors investigate the problem of influence maximization in presence of a competitor in the network. Also, they relax the constraint of discrete budget allocation, which is one of the most used assumptions in settings considered for the influence maximization problem. Specifically, the authors study the problem when different amounts may be allocated to different nodes, i.e., continuous influence maximization. First, they analyze the case when the competitor is passive, i.e., it has already influenced some of the nodes. Next, they also analyze two active opponents, using a game theoretical approach. Their results suggest that while degree-based methods are better suited for discrete budget allocations, in continuous influence maximization,

mimicking the targets of the opponent or selecting the direct neighbors of the opponent's targets are strategies superior to selecting hub nodes.

The paper is well written, and includes interesting results obtained with a sound methodology. Based on these considerations, I recommend the publication of the paper provided that the authors address the following issues.

[major]

1) Is it fair to make a straight comparison of performance between degree centrality and other optimization strategies? When using shadowing or shielding, the heuristics have the information on the moves of the opponent. However, when using the degree heuristic, such an information is not available to the method. This fact creates an obvious disadvantage for degree centrality. The observation does not aim at diminishing the importance of the results of the paper. However, the very fact that the information available to the methods is different should be clearly pointed out.

2) Why haven't the authors tried their methods on any other real networks? Additional tests, especially on networks with different size, are needed to understand the extent of the results of the paper.

3) Results are presented as they would be valid for very generic settings.

However, it should be clearly stated that the results of the paper are valid

for the competitive setting only. In particular, the disadvantages of the degree heuristic should be made more apparent than they are in the current version of the paper. Also, it should be remarked

that the presented results do not extend to the case where

only a single entity is responsible for network spreading.

[minor]

4) Line 161: Where did the constant 8 come from? Is it an arbitrary value selected on the basis of the information on the system available to the authors? If so, are there any other constants that make sense, and how do these constant values affect the results of node groupings?

5) Fig4: What does it happen when the active controller has a budget advantage? Also, in Fig.4c why aren't any points for N_b in the range x=3 to x=65? Is there an intuitive explanation for this behavior?

6) Ref. 43: The name of the first author should be Erkol Ş.

6. PLOS authors have the option to publish the peer review history of their article (what does this mean?). If published, this will include your full peer review and any attached files.

Reviewer #1: No

Reviewer #2: No

---

## [Author Response · Author response to Decision Letter 0]

12 Apr 2021

All responses to the editor and reviewers has been uploaded separately in a file named 'Response to Reviewers.pdf'.

---

## [Decision Letter · Decision Letter 1]

18 May 2021

Shadowing and shielding: Effective heuristics for continuous influence maximisation in the voting dynamics

PONE-D-21-02851R1

Dear Dr. Romero Moreno,

We’re pleased to inform you that your manuscript has been judged scientifically suitable for publication and will be formally accepted for publication once it meets all outstanding technical requirements.

Kind regards,

Hocine Cherifi

Academic Editor

PLOS ONE

Reviewers' comments:

Reviewer's Responses to Questions

**Comments to the Author**

1. If the authors have adequately addressed your comments raised in a previous round of review and you feel that this manuscript is now acceptable for publication, you may indicate that here to bypass the “Comments to the Author” section, enter your conflict of interest statement in the “Confidential to Editor” section, and submit your "Accept" recommendation.

Reviewer #1: All comments have been addressed

Reviewer #2: All comments have been addressed

2. Is the manuscript technically sound, and do the data support the conclusions?

Reviewer #1: Yes

Reviewer #2: Yes

3. Has the statistical analysis been performed appropriately and rigorously? 

Reviewer #1: Yes

Reviewer #2: Yes

4. Have the authors made all data underlying the findings in their manuscript fully available?

Reviewer #1: Yes

Reviewer #2: Yes

5. Is the manuscript presented in an intelligible fashion and written in standard English?

Reviewer #1: Yes

Reviewer #2: Yes

6. Review Comments to the Author

Reviewer #1: The authors have successfully addressed all of my comments.

Reviewer #2: The authors have properly addressed the comments provided by the reviewers. The manuscript can be accepted for publication.

7. PLOS authors have the option to publish the peer review history of their article (what does this mean?). If published, this will include your full peer review and any attached files.

Reviewer #1: No

Reviewer #2: No

---

## [Editor Report · Acceptance letter]

9 Jun 2021

PONE-D-21-02851R1 

Shadowing and shielding:  Effective heuristics for continuous influence maximisation in the voting dynamics 

Dear Dr. Romero Moreno:

I'm pleased to inform you that your manuscript has been deemed suitable for publication in PLOS ONE. Congratulations! Your manuscript is now with our production department. 

Kind regards, 

on behalf of

Professor Hocine Cherifi 

Academic Editor

PLOS ONE